# Early Alterations of Intra-Mural Elastic Lamellae Revealed by Synchrotron X-ray Micro-CT Exploration of Diabetic Aortas

**DOI:** 10.3390/ijms23063250

**Published:** 2022-03-17

**Authors:** Aïcha Ben Zemzem, Xiaowen Liang, Laetitia Vanalderwiert, Camille Bour, Béatrice Romier-Crouzet, Sébastien Blaise, Michael J. Sherratt, Timm Weitkamp, Manuel Dauchez, Stéphanie Baud, Nicolas Passat, Laurent Debelle, Sébastien Almagro

**Affiliations:** 1UMR MEDyC, CNRS 7369, Université de Reims Champagne Ardenne, SFR CAP SANTE, 51100 Reims, France; aicha.ben-zemzem@univ-reims.fr (A.B.Z.); xiaowen.liang@univ-reims.fr (X.L.); laetitia.vanalderwiert@univ-reims.fr (L.V.); camille.bour@univ-reims.fr (C.B.); beatrice.romier-crouzet@univ-reims.fr (B.R.-C.); sebastien.blaise@univ-reims.fr (S.B.); manuel.dauchez@univ-reims.fr (M.D.); stephanie.baud@univ-reims.fr (S.B.); 2CReSTIC, Université de Reims Champagne Ardenne, 51100 Reims, France; nicolas.passat@univ-reims.fr; 3Division of Cell Matrix Biology and Regenerative Medicine, School of Biological Sciences, Faculty of Biology Medicine and Health, The University of Manchester, Manchester M13 9PT, UK; michael.j.sherratt@manchester.ac.uk; 4Synchrotron SOLEIL, 91192 Gif-sur-Yvette, France; timm.weitkamp@synchrotron-soleil.fr

**Keywords:** arterial wall, elastic fibers, diabetes, X-ray microtomography, accelerated aging, extracellular matrix

## Abstract

Diabetes is a major concern of our society as it affects one person out of 11 around the world. Elastic fiber alterations due to diabetes increase the stiffness of large arteries, but the structural effects of these alterations are poorly known. To address this issue, we used synchrotron X-ray microcomputed tomography with in-line phase contrast to image in three dimensions C57Bl6J (control) and db/db (diabetic) mice with a resolution of 650 nm/voxel and a field size of 1.3 mm^3^. Having previously shown in younger WT and db/db mouse cohorts that elastic lamellae contain an internal supporting lattice, here we show that in older db/db mice the elastic lamellae lose this scaffold. We coupled this label-free method with automated image analysis to demonstrate that the elastic lamellae from the arterial wall are structurally altered and become 11% smoother (286,665 measurements). This alteration suggests a link between the loss of the 3D lattice-like network and the waviness of the elastic lamellae. Therefore, waviness measurement appears to be a measurable elasticity indicator and the 3D lattice-like network appears to be at the origin of the existence of this waviness. Both could be suitable indicators of the overall elasticity of the aorta.

## 1. Introduction

Diabetes is a chronic pathology affecting one person out of 11 around the world. It increases the probability of cardiovascular disease and accelerates arterial stiffness. The effect of diabetes on the stiffness of the vascular system has been broadly estimated to correspond to 6–15 years of chronological aging [1,2]. This pathology is mainly characterized by an important level of blood glucose leading to an overproduction of advanced glycation end products (AGEs). For example, hyperglycemic accumulation of AGEs in the vessel wall leads to the formation of cross-links within elastin fibers and collagens, resulting in a diminution of arterial wall distensibility [3]. The processes by which diabetes affects the arterial wall and especially elastic fibers are numerous [4] but the kinetics of the tissue-wide alterations remain unclear.

Large arteries such as the aorta contain, in their extracellular matrix (ECM), an amorphous elastic hydrophobic polymer named elastin, which is the major component of elastic fibers. Its entropic elasticity and its concentric lamellae organization within the aortic wall allow the aorta to remain in a reversible deformation domain during the cardiac cycle, which alternates phases of overpressure and contraction. Alterations to elastin due to a pathological situation (vascular complications of Marfan syndrome, aneurysm formation, etc.) or aging permanently affect the arterial mechanics [5]. As elastin has a half-life of 70 years [6] and an imperfect resynthesis after childhood, the detection of aberrant remodeling and its effects on tissue micro-structure is a major concern in understanding numerous pathologies.

Imaging of the arterial wall [7] is accessible in vivo by magnetic resonance imaging techniques or echography [8] but the resolution of these techniques is too low to characterize the structural details of intramural elastic structures. Optical coherence tomography has a better resolution (approx. 10 µm) but requires an arterial diameter of more than 5 mm and has a limited penetration. Higher resolution ex vivo techniques suffer from other technical problems. As aortic tissues are very elastic, imaging them is a challenge notably because these structures collapse after harvesting. Histology is commonly used to image biological tissues, but the sectioning procedure required to obtain the thin layers of tissue needed (3–5 µm thickness) may alter its fine structural organization. Fluorescence microscopy, including two-photon microscopy, is widely used notably because elastin is highly autofluorescent. However, fluorescence techniques are also limited by tissue thickness, ~500 and ~1300 µm for, respectively, 2- and 3-photon microscopy [9], and are therefore restricted in the size of the 3D observable field. Another limitation of conventional techniques, stressed by Kozel and Mecham [10], is the fact that elastin internal features are only maintained, and thus observable, when soft methods are used to prepare the samples.

In contrast, X-ray microtomography allows imaging of large sample volumes (imaging depth of several cm) with a micrometric to sub-micrometric resolution associated to a field size in the order of millimeters. Initially, Walton and colleagues [11] demonstrated the capability of this technique to image structures of rat carotid artery using phase contrast on a laboratory micro-CT machine. This technique was then improved by Lopez-Guimet and colleagues [12] who used synchrotron X-rays coupled to in-line phase contrast to enhance the image resolution (1.1 µm/voxel) and contrast of wild-type and Marfan syndrome mice. Recently [13], synchrotron X-ray micro-CT was used by our group to image millimeter-sized volumes of mouse aorta and surrounding tissues with an excellent signal-to-noise ratio at a better resolution of 812 nm/voxel, 35% better than Lopez-Guimet and colleagues. This technique, coupled for the first time to a fully automated segmentation procedure, revealed a hidden lattice-like filamentous structure in 3-month C57Bl6J (wild type) and db/db (diabetes model) mice. This filamentous structure, inside elastic lamellae of the arterial wall, presented an architecture similar to a shielded cable, seeming to reinforce the arterial wall so that it could manage blood pressure alterations.

In the present work, our goal was to analyze the effect of diabetes on the lattice-like network previously described and to address the effects of this pathology on arterial wall features. We imaged older mice, 6-month-old (C57Bl6J and db/db), to allow more time for the disease to manifest. We used in-line phase contrast synchrotron X-ray micro-CT with a resolution of 650 nm/voxel, improving image resolution by 25% in comparison of our previous work [13]. We created a new protocol of sample preparation using low-melting agarose injection in the lumen to apply a weak pressure and therefore avoid arterial collapse. We further developed a new image analysis approach to quantify the reserve length of the elastic lamella, an intrinsic property of elastic materials directly related to their elasticity. Using these approaches, we demonstrate that both scaffolding network and elastic lamellae are altered during the first steps of diabetes, indicating that these two parameters are suitable indicators of diabetes alteration and overall elasticity of the aorta.

## 2. Results

### 2.1. Typical Result of Synchrotron X-Ray Micro-Computed Tomography of a Mouse Artery

Figure 1a shows a typical reconstructed raw 2D image plane extracted from a synchrotron X-ray micro-CT acquisition (2048 × 2048 × 2048 voxels) for an aortic segment of a 6-month C57Bl6J mouse (control). The sample was not sectioned and no labeling or contrasting agents were used. The phase-contrast image exhibits the different structures of the aortic environment from the thoracic region. At a voxel size of 650 nm (corresponding to the effective pixel size of the detector), the aortic lumen (L), the wall (W), and the perivascular adipose tissue (P) are clearly visible. Enlarged areas in Figure 1a reveal the small details of these structures. After segmentation of the same region (Figure 1b) by a purpose-written fully automated program [13], the elements of the arterial wall appear clearly segmented from the supporting paraffin and agarose. At a larger scale, Figure 1c shows a 3D view of the full segmented volume stack (1.331 mm long, 2048 images) and the details of a secondary artery branching. The resolution of this technique, its signal-to-noise ratio, and the absence of contrasting agent is convenient for a detailed study of the structure of aortic intramural features.

### 2.2. Comparison to Diabetes Mouse Model

The C57Bl6J and db/db mouse models share the same genetic background and therefore differences between the two groups would not originate from genetic variation between strains but solely from the pathological situation. Figure 2 shows synchrotron X-ray micro-CT of aortic segments of control and diabetic mice. The aspect of the elastic lamellae (medial and intimal) exhibits drastic differences (Figure 2b,c). The control (upper row of images) presents a wavy aspect indicating that this elastic tissue still contains a reserve length. The reserve length of a spring, in addition to its stiffness, increases its resistance to stress under an acceptable constraint. This reserve length allows a spring to be stretched above its ‘apparent length’, i.e., its end-to-end distance. If a spring is stretched too much or altered, it loses its reserve length and thus loses most of its ability to be reversibly deformed again when overstretched. The consequence of losing reserve length is to increase the probability to undergo local plastic (i.e., irreversible) deformation when it will be exposed to the same constraint as before.

As can be seen in Figure 2, the elastic lamellae of the aortic wall of the 6-month-old diabetic model are smoother than those of the control at the same age, suggesting that these lamellae have lost a significant part of their reserve length and thus have their elasticity modified. Although a simple visual inspection of the tomography images clearly shows this smoother aspect, we decided to compute a measurable descriptor to quantify this wavy/smooth aspect between 21 tomographic volume scans (from 4 control mice and 6 diabetic mice).

### 2.3. Quantification of the Wavy/Smooth Aspect of the Elastic Lamellae of the Arterial Wall

One way to compare the wavy aspect to the smooth aspect is to compute, at the level of one curvy segment that comes from a single elastic lamella, the ratio (r_ec_) between its end-to-end distance and its curvilinear length (i.e., its real length) as visible in Figure 3a. For a wavy aspect, we have r_ec_ < 1 while for a perfectly straight line, we have r_ec_ = 1. Briefly, to gather segments of elastic lamellae, we thresholded each image of the stack, separated segments from other objects in the image, skeletonized the segments, and performed measurements. In this way, we were able to measure 15–30 segments per image (Figure 3b) and thus to obtain about 2000–30,000 measured segments per volume, from 300 to 900 tomographic slices in the volume. These segments were used to estimate an average ratio along all the elastic lamellae of a large-scale fragment of the aorta. In total, 127,180 measurements were made on control mice, 159,485 on diabetic mice.

The results obtained from these measurements of r_ec_ are presented in Figure 4. The comparison of the two groups C57Bl6J (mean = 0.823, sd = 0.047) versus db/db (mean = 0.889, sd = 0.026) indicates that r_ec_ has a highly significantly lower value in control as compared to diabetic mice (*p* < 0.004, Wilcoxon–Mann–Whitney two sample test). Hence, the reserve length of the control is significantly higher than the one of diabetic mice, indicating an important modification of elastic lamellae in pathological mice. When our analysis is limited to data from aortic regions, we reached the same conclusion about r_ec_ modification (*p* < 0.02) and this effect is not significant for the thoracic region (*p* < 0.065) but close to significance. Altogether these results indicate that diabetes tends to reduce the reserve length in a global way, this effect is more pronounced on the abdominal part.

### 2.4. Origin of the Loss of the Wavy Aspect of Elastic Lamellae

In the present work, mice were 6-month-old and the detailed observation of their lattice-like network exhibits clear differences between control and diabetic mice. As can be seen in Figure 5 and Video S1, the lattice-like network is readily visible in control mice, while it has almost entirely disappeared in diabetic model samples, whatever the threshold value is. Our imaging technique relies on phase-contrast and therefore is sensitive to refractive index. Thus, this observation indicates that the local composition and/or 3D organization of the elastic structures has changed. It is possible that the material remains in its place, but its internal organization is altered.

## 3. Discussion

In-line phase-contrast synchrotron X-ray microtomography of the arterial wall with label-free tissues grants access to fine details of elastic structures. The strength of this method lies in its large field, its excellent resolution, and a good contrast for intramural elastic structures. Coupled to an automatic image analysis program, this technique provides information that cannot be obtained using other imaging methods (Figure 1 and Figure 5). Further, for each sample, it allows the computation of thousands of measurements for a given feature, such as the ratio between Euclidean and geodesic distances of a segment of an elastic lamella (Figure 3 and Figure 4).

Our results show that the reserve length is significantly reduced in diabetic mice together with the filaments that support the wall. This suggests that elasticity is correlated to the existence of this lattice-like network that has disappeared at 6 months of diabetes.

The loss of arterial pressurization is commonly known to be at the origin of the wavy aspect of the ex vivo artery segment. Nevertheless, the waves, which reflect the reserve length, also exist on the intimal elastic lamella at physiological pressure as seen on Appendix A. Our sample preparation process used agarose injection to maintain a weak counterpressure inside the aorta so as to avoid its collapse after harvesting. Under these conditions, waviness disappears from diabetic mice but is still visible in control mice (Figure 5).

As elastic fibers from the vessel wall appear amorphous with conventional transmission electron microscopy, they have long been considered as devoid of internal organization, meaning that their alterations would be diffuse along their fabric. Recently, Kozel and Mecham [10] demonstrated, using deep etch images of growing elastic fibers, that elastin is composed of a densely packed and disordered arrangement of filamentous substructures. Their study also indicates that these structures remain intact using soft and label-free techniques.

We have recently shown that elastic lamellae of the mouse arterial wall contain a lattice-like network of filaments of an unknown composition [13]. This network was visible either in 3-month-old C57Bl6J (control) or db/db (diabetes model) mice. The fact that diabetes alters the arterial wall has been known for decades, but the nature of these modifications is still poorly understood but has been demonstrated to be correlated with elastic lamellae alterations [5]. We here observe that this network has nearly completely disappeared from the arterial wall of 6-month-old diabetic mice but not from control mice (Figure 5). This disappearance is accompanied by the loss of the wavy aspect of elastic lamellae in 2D cross-sectioning view (Figure 2 and Figure 4).

The lattice-like network filaments could be the consequence of a visual 3D effect. We, therefore, checked their presence in 2D orthogonal views. They are also visible (Appendix A) rejecting totally this possible bias. As we have demonstrated that neither the lattice-like network nor the wavy aspect of intramural features are artifactual, we suggest that the wavy aspect, and so the elasticity of the arterial wall, is linked to the presence of these filaments.

As a truism, the wavy aspect of the harvested arteries has always been imagined as a ‘random contraction’ of the elastic structures of the arterial wall. However, an elastic element such as a rubber band, does not present directed and localized ripples, such as the ones observed for a collapsed aorta, in the absence of stress. Our data suggest that even when pressurized arterial lamellae have a wavy internal sub-structure which then drives the waviness of the lamellae once pressure is removed. We therefore propose that these ripples are directly linked to the existence of these lattice-like filaments. From a mechanical point of view, these ripples could reflect the ability of the vascular wall to inflate and retract after stress. Losing these filaments will certainly affect the stiffness of the wall. Physiologists working with db/db mice observe the great fragility of their aorta especially during harvesting steps. This suggests that the microscopic effect we observed could have macroscopic consequences on arterial resilience. As the disappearance of filaments is the most measurable and visible difference we observe between the two groups, we suggest that damaging these filaments could be one of the first alterations induced by diabetes. Therefore, we propose that the wavy aspect of the elastic lamellae is a measurable indicator of its elastic integrity and the 3D lattice-like network inside elastic lamellae is at the origin of the waviness. Both could be suitable indicators of the overall elasticity of the aorta.

The results obtained by our method provide an original vision of the architecture of vascular tissues in relation to their mechanics. Even if this synchrotron X-ray microtomography technology is not directly transferable to living humans due to the radiation dose, the results it provides will allow future developments to evaluate the presence of these filaments in vivo, probably using OCT. Indeed, as the human elastic aorta wall is thicker and contains more elastic lamellae, the global persistence length is higher leading to larger curvature deformation that could potentially be detectable through the improvements of this technique.

## 4. Materials and Methods

### 4.1. Animals

Mouse procedures were realized in accordance with the Guide for the Care and Use of Laboratory Animals of the US National Institutes of Health and were authorized by the Animal Subjects Committee of the Champagne-Ardenne region. The mice, 6-month-old C57BL6J (control, *n* = 4) and 6-month-old db/db (diabetic, *n* = 6), were purchased from Charles River (Lyon, France). They were caged in a 12:12 h light/dark cycle in a temperature and humidity-controlled environment.

### 4.2. Sample Preparations

Aortas were collected as follows. After euthanasia, 2500 UI heparin (Sanofi, Paris, France) were injected in the heart to prevent blood coagulation. The heart and aorta were then flushed with 10 mL phosphate buffered saline using a syringe driver at 2 mL/min for control mice and 1.5 mL/min for diabetic mice (Harvard Apparatus, Holliston, MA, USA) to remove residual blood. The aorta was further prefixed by injecting 5 mL of 4% formalin with the same flow. Then, 5 min after the end of the fixative solution injection, 6 mL of 1% low melting agarose (Fisher Bioreagents, Waltham, MA, USA, BP165-25) were injected (2 mL/min control, 1.5 mL/min diabetic mice) to keep the aorta opened and to prevent collapse during dehydration and inclusion steps. The system was then left to rest at room temperature for 5 min so that the agarose solidified as a gel in the lumen and exerted a counterpressure to avoid collapse. The heart and aorta were collected with surrounding tissues and placed in a cassette where the aorta was held at both ends by a wire to keep it straight. Finally, the samples were fixed in 4% formalin for 24–48 h, dehydrated, and embedded in paraffin. The final samples were about 4-cm-long paraffin rods containing the heart and aorta.

### 4.3. Imaging Procedures

Synchrotron X-ray microtomography was performed on the ANATOMIX beamline at the SOLEIL synchrotron (Gif-sur-Yvette, France) [14]. Samples were imaged with a polychromatic (“white”) X-ray beam obtained from an undulator X-ray source set to a gap of 8.5 mm; the beam was filtered by a 0.6-mm-thick diamond plate and a 10-µm-thick layer of gold. The detector was an indirect lens-coupled system with a 20-µm-thick lutetium aluminum garnet single-crystal scintillator (Crytur, Turnov, Czech Republic) coupled to a CMOS-based scientific-grade camera (Orca Flash 4.0 V2, Hamamatsu, Japan) with 2048 × 2048 pixels via microscope optics (10x objective, Mitutoyo, Japan), resulting in an effective pixel size of 0.65 µm on the sample level. The distance between sample and scintillator was 22 mm. The exposure time for the camera was set to 100 ms per projection image. In total, 1500 projections were taken over an angular range of 180°. The samples were positioned vertically with the heart in the upper part. Three scans were performed on each sample: one near the aortic arch (descending aorta close to the heart), one in the thoracic zone and one in lower part, say in the abdominal zone. Immediately after the acquisition, the imaged volume was reconstructed and assessed before saving the data. Tomographic reconstruction was performed using the standard processing pipeline at the beamline, based on a Python script and subsequent reconstruction by the PyHST2 software [14]. The reconstructed volume stacks for each scan contained 2048 × 2048 × 2048 voxels of size 0.65 µm^3^, each represented by a 32-bit single-precision float value, i.e., a total of 32 GB per stack. After reconstruction, we manually controlled scans, and we eliminated invalid acquisitions. Bad scans were removed most of the time because of the presence of artifacts such as air bubbles, strong deformed shape of the arteries, arteries damaged during preparation or handling, or poor contrast.

### 4.4. Software Used for Visualization and Analysis

Basic image and video manipulations were performed using ImageJ (National Institutes of Health, Bethesda, MD, USA) for 3D volumetric visualization figures and the ImageJ 3D Viewer plugin for videos. Figures were created with Adode Photoshop CC 2014, Adobe Illustrator CC 2014 and Video editing was completed with Adobe After Effects CC 2014 (San José, CA, USA). Our image analysis software was scripted with Matlab r2021a (MathWorks, Natick, MA, USA) including the VP/CF/IP/DM/SG/ST toolboxes.

### 4.5. Arterial Structure Segmentation

The computer used was a dual processor Intel Xeon E5 2680 V2 (2 × 10 cores) with 128 GB RAM, high speed SSD disks, and a NVIDIA GeForce 2080Ti with 11GB of RAM. Arterial structures were segmented automatically from reconstructed stacks with bespoke Matlab scripts as previously described [13]. The result of this segmentation for each tomography volume scan is a 2048-image 3D stack containing 8-bit voxels of the arterial wall. Note that this program implements a segmentation method and does not affect the voxel intensity.

### 4.6. Measurements of the Reserve Length of Elastic Lamellae

#### 4.6.1. Creation of The Image Database of Elastic Lamellae Fragments

Due to their structure, elastic lamellae of the arterial wall could be very close even at a sub-micrometric level or connected. As they appear as curves on 2D micro-CT images, we first thresholded above about 2 times the value obtained by the Otsu’s method [15]. Each region of the image containing the signal of a fragment of an elastic lamella was then measured to only keep the regions that (1) presented an area above or equal to 100 pixels (in order to discard the smallest regions), (2) had a ratio below 15% of the pixels of the bounding box occupied by the elastic lamellae (in order to delete part of the image were elastic lamellae are superimposed), and (3) presented a ratio of less than 36% of the pixels in the convex hull that are also in the region (in order to eliminate unstructured zones as the external adventitia). In a third step, the skeleton of each kept fragment was computed and only the longest path of the skeleton was conserved. Typically, about 15–30 fragments are kept per image, leading to about 2000–30,000 fragments from 300–900 images, as we restrained the number of images due to computing time. We manually checked that this process kept fragments of elastic lamellae which are a faithful reflection of each sample.

#### 4.6.2. Deducing the Reserve Length from Each Image

We assumed that (1) a skeleton curve, composed of pixels is a digital approximation of the underlying continuous curve/lamella cross-section trajectory and (2) the polygonal sampling of such a continuous curve provides scaled estimation of its length, from Euclidean-like distance (for coarse sampling) to geodesic distance (for fine sampling). Based on these facts, for a given sampling parameter factor s (s = 1 means that we use the initial pixels to compose curves, s = 2 means that we select every other pixel of the initial ones, etc.), we computed the polygonal line composed of points of the curve sampled every s pixels (here we used 3 pixels), and we then oversampled this discrete curve regularly in order to obtain a polygon finally composed by a number of vertices equal to the initial number of pixels by inserting (s − 1) vertices between each pair of adjacent vertices on the discrete curve. Two distances were then estimated from this polygonal line. First, the Euclidean-like distance was computed as the sum of the Euclidean distances between the points sampled with a step of 100 units of geodesic-like length (or, for shorter polygonal lines between the first/last vertices or the first/middle + middle/last vertices where the middle vertex refers to the one corresponding to half of the geodesic-like length). This first distance corresponds to the end-to-end distance (green line) in Figure 3a,b. Second, the geodesic-like distance (i.e., the curvilinear length) was computed as the sum of the distances between the successive vertices of the polygonal line (these vertices may be subpixels as s = 3). The ratio between the end-to-end and the curvilinear length distances then corresponds to the r_ec_ ratio provided in Figure 4.

### 4.7. Statistical Analysis

Samples harvested from animals were organized in two groups: control from C57BL/6J mice (*n* = 4) and diabetic from db/db mice (*n* = 6). All animals were aged 6 months. Prior to the analytical computation, image stacks presenting artefacts that could bias the analysis were discarded. Finally, 10 images stacks were retained for control mice (127,180 images) and 11 images stacks from diabetic mice were conserved (159,485 images).

Statistical analyses were performed using the R software version 4.1.1 (R Foundation for Statistical Computing, Vienna, Austria) implemented in the RStudio IDE version 2021.09.2 (RStudio Inc., Boston, MA, USA). P-values were computed using the non-parametric Mann–Whitney–Wilcoxon rank sum test with continuity correction. The choice of this test was due to (1) the sample number (Figure 4), (2) a computed data normality for db/db samples close to the significance level (*p*-value = 0.079) as assessed by the Shapiro–Wilk test, and (3) not enough satisfactory normal quantile-quantile plots. We verified that modifying the s value (with 1, 2, or 3) for computing r_ec_ does not affect the global result of the statistical tests.

## Figures and Tables

**Figure 1 ijms-23-03250-f001:**
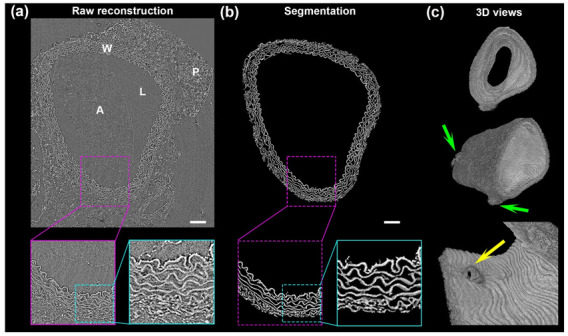
Typical result of a synchrotron X-ray microtomography acquisition with in-line phase-contrast of a paraffin-embedded 6-month C57Bl6J (control) thoracic aortic segment. The voxel size is 0.65 µm and all images are from the same acquisition. (**a**) One 2D raw reconstruction plane shows the lumen (L) of the artery surrounded by the arterial wall (W) itself surrounded by the perivascular adipose tissue (P). Some agarose (A) remains in the lumen after sample preparation. Successive magnification steps (purple and cyan) reveal details of the plane. (**b**) Results of the same areas after segmentation of the elastic structures and magnifications (purple and cyan) demonstrate the precision of the automated segmentation. (**c**) 3D views of a full volume reconstruction (2048 × 2048 × 2048 voxels). Front view (**top**) clearly shows the luminal face, three-quarters view (**middle**) shows 1.33 mm of the reconstruction with two secondary artery branching (green arrows). Inside view (**bottom**) of a secondary artery connection (yellow arrow). Scale bars = 100 µm.

**Figure 2 ijms-23-03250-f002:**
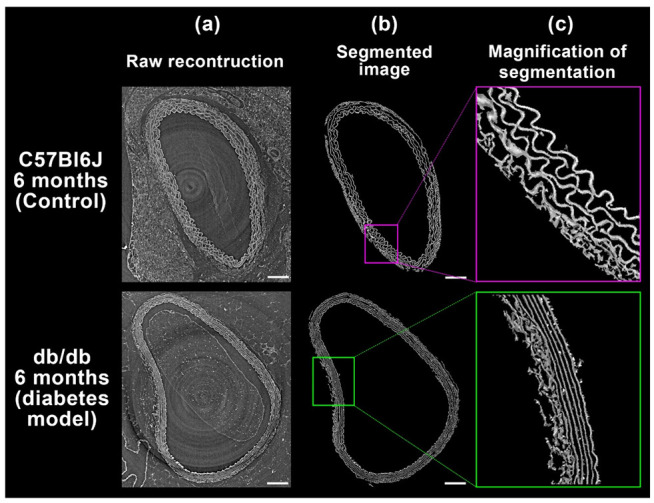
Effect of diabetes on the aortic structure of the mouse arterial wall. Typical result of synchrotron X-ray microtomography acquisition (650 nm/voxel) with in-line phase-contrast of a paraffin-embedded 6-month mouse aorta segment. Raw reconstruction (**a**) and segmented image (**b**) are shown. Control mouse (upper images row) and diabetes mouse models (lower images row) exhibit significant differences in the appearance of their elastic lamellae as shown by enlarged details (**c**) respectively framed in purple and green, that reveal wavy lamella structures for the control specimen and smooth ones for the diabetic mouse. Scale bars = 100 µm.

**Figure 3 ijms-23-03250-f003:**
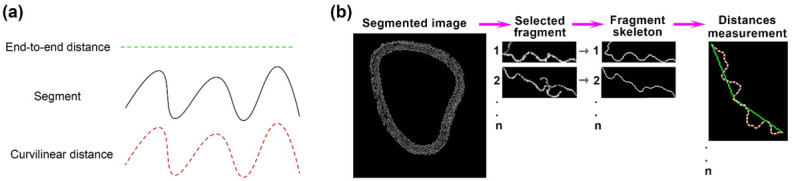
Measurement of the wavy/smooth aspect of the elastic lamellae. (**a**) Scheme illustrating the difference between the end-to-end distance (dashed green line) and the curvilinear distance (dashed red line) for a given segment (black line). (**b**) General view of the process of the quantification of the wavy aspect. Purple arrows indicate the direction of the workflow. First, each segmented image of the stack is thresholded to select individual fragments (1 to n). Each fragment is then skeletonized in a single fragment with two ends. In the rightmost image, each skeleton is subsampled every 3 pixels to reduce pixel discretization error, the curvilinear distance is calculated based on these subsampled points (red), the green line indicates the end-to-end distance.

**Figure 4 ijms-23-03250-f004:**
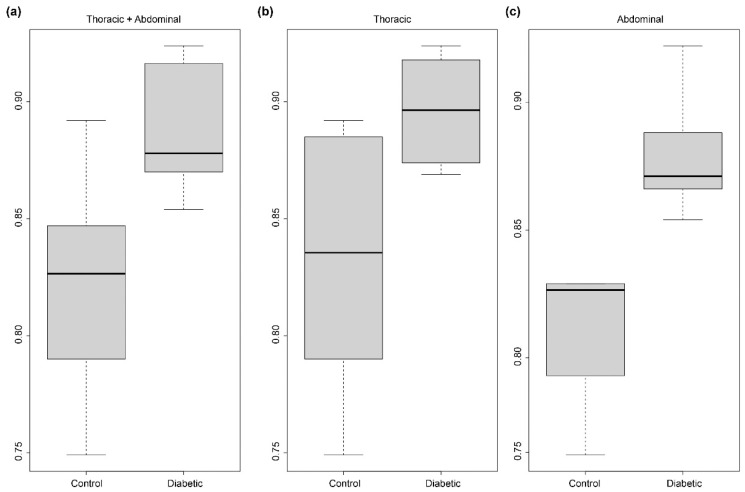
Box charts of ratio r_ec_ obtained for control (C57Bl6J) and diabetic (db/db) 6-month mice. (**a**) All regions together (*p*-value < 0.004), (**b**) only the thoracic region (*p*-value < 0.065), and (**c**) only the abdominal region (*p*-value < 0.02).

**Figure 5 ijms-23-03250-f005:**
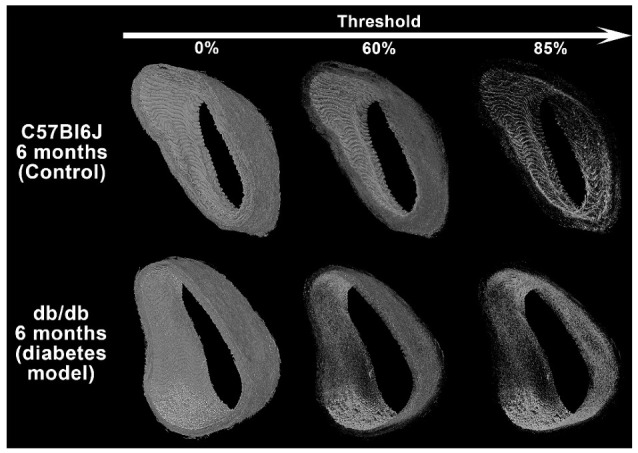
Lattice-like filamentous networks of 6-month-old mice. Typical 3D reconstruction of synchrotron X-ray micro-CT acquisition of C57Bl6J mouse (upper images row) and db/db mice (lower images row). Stacks are limited to 800 planes for convenient visualization. To compare structural details, the samples are the same as in Figure 2. Voxel intensity threshold increases from left to right to reveal the underlying lattice-like network (C57Bl6J, upper line) or its absence (db/db, lower line). Largest lumen diameter: 670 µm (C57Bl6J) and, respectively, 730 µm (db/db).

## Data Availability

The data presented in this study are available on request from the corresponding author until year 2027 due to their large volume. Scripts are available on request from S. Almagro.

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
