# Peer review of "Early Alterations of Intra-Mural Elastic Lamellae Revealed by Synchrotron X-ray Micro-CT Exploration of Diabetic Aortas"

_ijms, 2022, doi:10.3390/ijms23063250_

Round 1
Reviewer 1 Report
The authors have revised in-depth their manuscript, which now is substantially improved.
No further actions are required.
Author Response
We wish to thank Reviewer 1 for his comments and suggestions.
Reviewer 2 Report
The manuscript " Early Alteratios of Intra-Mural Elastic Lamellae Revealed by Synchrotron X-Ray Micro-CT Exploration of Diabetic Aortas" is an interesting paper dealing with the potential effects of diabetes on aortic wall and the possibility of highlighting them by using synchrotron X-ray microcomputed tomography with in-line phase contrast to image in three dimensions. However, there are some issues that require further highlighiting:
1- I would like to ask Authors whether the Micro-CT technology is really ready to be transferred for the same purposes in human diagnostics. If yes,what's the radiation burden of this procedure? I suggest adding a brief comment on this.
2- Did Authors correlate the structural alterations caused by diabetes on the aortic wall with the degree of loss of elasticity of the aorta itself? I suggest adding a brief comment on this.
Author Response
We wish to thank Reviewer 2 for his comments and suggestions.
To answer to the first question, a paragraph has been added at the end of the discussion to address this point, see p8 lines 272 to 279.
To answer to the second question, this point is very interesting because the elasticity of the aorta itself, i.e. its resilience of the entire object, is the macroscopic translation of the micrometric effect measured. We were not able to measure the global elasticity of the aorta because aorta from 6-month-old db/db mice are very fragile in comparison to C57Bl6J samples. Therefore, we always perform a prefixation step before aorta harvesting which prevents us from making the direct measurement of elasticity. But, as indirectly suggested by Reviewer 2, the resilience (i.e. the fragility) of the diabetic aorta is visible experimentally when the samples are prepared. We observed this phenomenon, but we could not quantify it. The text has been modified to clarify this point: see p8-lines 264 to 266.
Reviewer 3 Report
Overall, this is a nicely written manuscript reporting the use of synchrotron micro-CT in the assessment of elastic fiber changes in mic aortas. Results advance our understanding of the impact of diabetes on vascular network alterations.
Specific comments:
- Introduction: page 2, 2nd para, authors stated the OCT has low resolution. I disagree as OCT has superior resolution of 20 um, although it has limited penetration of only 2cm. Please revise it.
- Results: page 6, 1st para under the Origin of the loss of the wavy aspect of elastic lamellae, this paragraph is more like discussion instead of presenting results as authors discussed findings in relation to other studies. It should be moved to the Discussion. Please consider it.
Author Response
We wish to thank Reviewer 3 for his comments and suggestions.
To answer to the first comment, we have modified p2 lines 56 to 59 to address this point.
To answer to the second comment, the paragraph has been deleted (p6 lines 192-198) and its contents have been inserted in the discussion (p7-8, lines 242-245).